



# HIPPO environmental monitoring: Impact of phytoplankton dynamics on water column chemistry and the sclerochronology of the king scallop (*Pecten maximus*) as a biogenic archive for past primary production reconstructions

Valentin Siebert[1], Brivaëla Moriceau[1], Lukas Fröhlich[2], Bernd R. Schöne[2], Erwan Amice[1], Beatriz Becker[4], Kevin Bihannic[1], Isabelle Bihannic[1], Gaspard Delebecq[1], Jérémy Devesa[1], Morgane Gallinari[1], Yoan Germain[3], Emilie Grossteffan[4], Klaus Peter Jochum[5], Thierry Le Bec[4], Manon Le Goff[1], Céline Liorzou[6], Aude Leynaert[1], Claudie Marec[5], Marc Picheral[7], Peggy Rimmelin-Maury[4], Marie-Laure Rouget[4], Matthieu Waeles[1], and Julien Thébault[1]

[1]Univ Brest, CNRS, IRD, Ifremer, LEMAR, F-29280 Plouzané, France
[2]nstitute of Geosciences, University of Mainz, Johann-Joachim-Becher-Weg 21, 55128 Mainz, Germany
[3]Department of Marine Geoscience, Institut Français de Recherche pour l'Exploitation de la Mer, Centre de Brest, 29280 Plouzané, France
[4]Institut Universitaire Européen de la Mer, IUEM, UAR 3113, Université de Bretagne Occidentale, CNRS, IRD , PLouzané, France
[5]Climate Geochemistry Department, Max Planck Institute for Chemistry, Mainz, Germany
[6]Université de Bretagne Occidentale, Laboratoire Geo-Ocean, CNRS-UBO-Ifremer-UBS, IUEM, Plouzané, France
[7]Sorbonne Université, Centre National de la Recherche Scientifique, Laboratoire d'Océanographie de Villefranche (LOV), Villefranche-sur-Mer, France

**Correspondence:** Valentin Siebert (valentin.siebert@univ-brest.fr)

**Abstract.** As part of the HIPPO (HIgh-resolution Primary Production multi-prOxy archives) project, an environmental monitoring was carried out between March and October 2021 in the Bay of Brest. The aim of this survey was to better understand the processes which drive the incorporation of chemical elements into scallop shells and their links with phytoplankton dynamics. For this purpose, biological samples (scallops and phytoplankton) as well as water samples were collected in order to analyse

5 various environmental parameters (element chemical properties, nutrients, chlorophyll, etc.). Here, some of the monitoring data are presented and discussed. The whole dataset is much larger and can potentially be very useful for other scientists performing sclerochronological investigations, studying biogeochemical cycles or conducting various ecological research projects.

## 1 Introduction

Primary producers form the foundation of marine food webs, control population sizes at higher trophic levels and influence

10 fish stock recruitment (Chassot at al., 2010). Marine photoautotrophic organisms are responsible for nearly half of the global net primary production and fix substantial amounts of carbon (Field et al., 1998). Despite their key role in the functioning of marine ecosystems and the global climate, past primary production dynamics and mechanisms controlling them have not been well characterized (Boyce et al., 2010), especially for time intervals prior to the instrumental recording and significant





anthropological disturbances of biogeochemical cycles. Available data sources on changes in marine primary production do not provide the necessary temporal resolution, are too short to determine long-term trends and do not cover the entirety of photoautotroph taxa. To compensate for this lack of past data, biogenic archives have been interrogated, such as bivalve shells, to track the past dynamics of primary production. It has already been shown that some bivalve species incorporate trace elements into their skeletal structures (e.g., calcite or aragonite) and that the observed trends follow the patterns of the dynamics of primary producers in the water column (e.g., Stecher et al., 1996; Barats et al., 2009; Thébault et al., 2009; Doré et al., 2020). In particular, the king scallop, *Pecten maximus* can record these signals on a daily scale, allowing high-resolution temporal reconstructions (Chauvaud et al., 1998; Thébault and Chauvaud, 2013; Fröhlich et al., 2022b). The research project HIPPO aims to better understand these environmental proxies and to develop new tracers and integrate them in a multi-proxy approach. For this purpose, environmental monitoring with a high temporal resolution was set up in 2021 in the Bay of Brest (France). Here, an overview of the physical, biological and chemical parameters of the seawater is presented along with geochemical data of king scallop shells. All data were obtained during the HIPPO survey.

## 2 Study site and sampling strategy

The Bay of Brest is a semi-enclosed, macrotidal, marine ecosystem of 180 km$^2$ with a westward connection to the Iroise Sea via a deep (40 m max. depth) and narrow (2 km width) strait. Freshwater inputs are brought from the east via two main rivers, the Aulne in the south and the Elorn in the north of the bay (Fig. 1A). The study was carried out between early March and mid-October 2021, with a total of 38 cruises off Lanvéoc (48°17'39"N – 004°27'12"W), a site located in the southern part of the Bay of Brest (France, Fig. 1A). This site was chosen for its status as a no-fishing and no-hunting zone, allowing for undisturbed monitoring. Moreover, this site has already been the subject of scientific surveys in the past, conducted by the LEMAR-Laboratory and the observatory service of the European Institute for Marine Studies (IUEM). Lanvéoc is characterized by shallow waters (from approx. 8 m to 15 m depending on the tide, with a mean depth of approx. 11.5 m recorded at mid-tide) and a seafloor made up of sandy and silty sediment, with significant amounts of large-sized biogenic detritus (shells, maerl).

Water samples from the surface (1 m deep) and bottom (approx. 20 cm above the sediment) were collected twice a week between the 16 March and the end of June 2021, and once a week thereafter until mid-October 2021. No sampling was conducted during August, leading to a lack of data for this month. Several biological, chemicals or physical parameters were measured in these seawater samples: an exhaustive list is given in Table 1. For samples dedicated to trace element measurements, a 5 L Teflon-coated GoFlo sampling bottle was used. For the measurements of all other parameters, seawater was collected in a 5 L or a 8 L Niskin bottle. In addition to seawater sampling, two NKE/Sambat probes were alternately deployed to track high frequency physical parameters (CTDO2FluoTurbpH – measurements every 20 minutes). For example, tidal dynamics were assessed in the Bay of Brest by recording the water depth at Lanvéoc (Fig. 1B). The two major episodes of near-maximum tidal range occurred around 30 March (tidal range: 7.15 m) and 28 April (tidal range: 7.00 m). Two other instruments were deployed to track particle dynamics and chemistry: an imaging sensor, the Underwater Vision Profiler (UVP6), and a sediment trap. The settings used, specific to each instrument, will be described further below.



In addition to seawater sampling, 2,640 age-class I specimens of the king scallop, *Pecten maximus* (whose growth ceases only during the winter season), were placed on the sediment surface on 21 February 2021, before the beginning of the HIPPO monitoring phase. These individuals come from the Tinduff hatchery (located in Plougastel-Daoulas, in the eastern part of the Bay of Brest) and are derived from a very limited number of spawners, and are thus genetically closely related. To assess the impact of the sediment on the incorporation of the different trace elements into their shells (calcite), 360 specimens were placed in a cage located 1 m above the seafloor, devoid of any sediment. During each cruise, five shells were collected directly on the sediment (when possible) and five others from the cage to analyze the soft tissues (gills, mantle, digestive glands and muscle) for element chemical composition. At the end of the monitoring interval (on 11 October 2021), a larger number of bivalves was collected to establish a growth and element chemical time-series for 2021.

## 3 Data overview

### 3.1 Physical data

To monitor some of the physical and chemical parameters in the water column, two NKE/Sambat probes were alternately deployed at the seafloor at Lanvéoc. Throughout the year, they recorded water depth (Fig. 1B), temperature, salinity, pH, fluorescence and dissolved oxygen concentration (DO) every 20 minutes. Figure 2 shows the signals obtained for each parameter after removing outliers and data windows on which sensors showed malfunctions or recorded outliers, hence the lack of data for some variables for some time intervals. Water temperature increased from approx. 10 °C in March to 18 °C in August, remaining stable at this value until the end of September before decreasing slightly until the end of the monitoring. Simultaneously, salinity increased more rapidly from March (approx. 32 psu) to May (approx. 34.3 psu), caused by lower freshwater inflow from the Aulne due to a gradual decrease of its flow rate during spring (Aulne flow data are available on HydroFrance website: https://www.hydro.eaufrance.fr). Afterward, the salinity remained stable around a value of 34.5 psu until 14 October. While a gradual increase of DO (8.2 to 11.2 mg.L$^{-1}$) and pH (7.96 to 8.22) was detected between January and mid-April/early May, chlorophyll concentration (deducted from fluorescence) showed a more abrupt increase during the first two weeks of April from 2.5 to 9.44 µg.L$^{-1}$. This time interval coincides with the first increase in phytoplankton cell concentration (see Sect. 3.3.1.) and was followed by a significant decrease in the recorded values for DO, chlorophyll and pH, which ended in late May, just before the onset of the main diatom bloom of the year (see Sect. 3.3.1.). These decreases in DO and pH values were probably related to the end of the spring bloom, when the respiration activity in the water column intensified. The increased consumption of oxygen was accompanied by a release of $CO_2$, which contributed to a decrease in pH. For the rest of the year, DO and pH remained at a stable value that oscillated around 8.5 mg.L$^{-1}$ and 8.05 respectively. However, fluorescence showed a flat background interrupted by sharp, narrow peaks, especially in June when the main diatom bloom was recorded, suggesting higher photosynthesis activity during this time of the year.

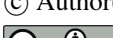



## 3.2 Nutrients

Concentrations of ammonium ($NH_4^+$), nitrate ($NO_3^-$), nitrite ($NO_2^-$), silicate ($SiOH_4$) and phosphate ($PO_4^{3-}$) were measured during the HIPPO monitoring scheme. For $NH_4^+$ concentration measurements, 100 mL glass vials were filled with seawater and immediately stored at -20 °C. To measure the other nutrients, 1 L of water was filtered through a 0.6 μm polycarbonate filter (Merck) within 3 h after sampling. The filtrate was recovered on 15 mL polypropylene tubes and stored at 4 °C for $SiOH_4$ measurements; the remainder was stored at -20 °C for $NO_3^-$, $NO_2^-$ and $PO_4^{3-}$ analyses. Then, the concentrations of all nutrients were measured with an AA3 HR SEAL-BRAN+LUEBBE auto-analyzer following the colorimetric (for $NO_3^-$, $NO_2^-$, $SiOH_4$ and $PO_4^{3-}$) or fluorimetric (for $NH_4^+$) methods of Amino and Kérouel (2007). Concentrations of $SiOH_4$, $PO_4^{3-}$, and dissolved inorganic nitrogen (DIN) (measured as the sum of $NH_4^+$, $NO_3^-$ and $NO_2^-$) reached their maximum on 07 February at sea surface (Fig. 3). Times of low nutrient concentrations were detected based on the half-saturation constant for nutrient uptake ($Km$), since nutrient concentrations below this constant correspond to stressful conditions that limit phytoplankton development (Del Amo et al., 1997). In the Bay of Brest, these constants equal 2.0 μmol.L$^{-1}$ for DIN and $SiOH_4$ as well as 0.2 μmol.L$^{-1}$ for $PO_4^{3-}$ (Del Amo et al., 1997).

Based on this method, $PO_4^{3-}$ proved to be the first limiting nutrient, with concentrations that dropped below the $Km$ as of 25 March, corresponding to the timing of the first increase in phytoplankton cells of the year (see Sect. 3.3.1.). $SiOH_4$ was the second limiting nutrient and had the shortest limitation interval (from 06 April to 21 June), which was interrupted by a slight enrichment in silicate at the end of May. Thereafter, silicate concentrations gradually increased until the end of the monitoring episode. DIN fell to limiting values three weeks after $PO_4^{3-}$ values became limiting, lasting from mid-April until mid-September. At the end of September, DIN and $PO_4^{3-}$ concentrations began to rise until the end of the experiment (Fig. 3).

## 3.3 Phytoplankton and organic matter dynamics

### 3.3.1 Phytoplankton taxonomy and counting

For phytoplankton determination and counts, a 250 mL glass bottle was carefully filled with the surface water through a silicon tube to avoid turbulence and disintegration of phytoplankton cells. Phytoplankton cells were immediately fixed in 2 mL of acid Lugol's iodine solution and stored in a dark and cold (4 °C) place to avoid UV damage and evaporation. The identification of phytoplankton taxa as well as the calculation of the respective cell concentrations were performed within a month of sample collection using an aliquot of 50 mL of the water sample, used to fill a sedimentation column combined with a plate chamber (Hydrobios Kiel), according the the Utermöhl methods (Edler and Elbrächter, 2010). After 24h, the phytoplankton cells settled on a glass microscope slide and were identified and counted using an inverted microscope (Axio Observer.A1-ZEISS).

Phytoplankton communities at Lanvéoc were predominantly composed of diatom species, which represented 87% of the counted cells in the total phytoplankton samples from 2021 (Fig. 4). Although the most dominant diatoms in the Bay of Brest are usually *Chaetoceros spp.* (e.g., Lorrain et al., 2000; Thébault et al., 2022), the two main blooms of 2021 mainly consisted of *Leptocylindrus danicus*, which occurred on 03 June (68% of total phytoplankton) and 15 September (95% of total phytoplankton). A smaller increase in the number of microalgal cells involving other species was also observed in mid-spring, caused by





the presence of *Cerataulina pelagica* and *Guinardia spp.* cells (Fig. 4A). This minor increase was particularly visible considering the biovolume of the species present, calculated with the equations provided in Smayda (1965) (Fig. 4B). A succession of several diatom blooms was observed which were composed of different species, such as *Coscinodiscus waileisii*, *Cerataulina pelagica* and *Guinardia flaccida*, which reached maximum abundances on 23 March, 20 April and 18 May, respectively, while *L. danicus* remained the dominant diatom species (Fig. 4B). Dinoflagellates were mainly represented by three species, i.e.,

*Gymnodinium sp.*, which showed three blooms on 04 May, 17 July and 15 September, as well as *Protoperidinium bipes* and *Prorocentrum triestinum*, which bloomed on 02 June and on 15 September, respectively (Fig. 4C). Potential causes for bloom decay will be given in Sect 3.3.2.

### 3.3.2 Organic matter analyses

Seawater samples dedicated to chlorophyll a (Chl), pheophytin (Pheo), particulate organic carbon and nitrogen (POC and PON,

respectively), and biogenic silica (BSi) analyses were collected from the surface and bottom waters using a 5 L or a 8 L Niskin bottle. Some samples for the bottom layer were collected using a syringe for a few dates (collecting strategy is specified in the dataset). Water samples were filtered through glass fiber filters (GF/F Whatman) pre-combusted for 4 h in a muffle furnace at 450 °C for Chl, Pheo, POC and PON measurements. BSi samples were analyzed after filtering seawater through 0.6 μm polycarbonate (Merck) filters. For each parameter, the filters were stored in an appropriate setting, i.e., POC/PON and BSi

samples in dark and dry place, Chl and Pheo samples at -80 °C. POC and PON concentrations were measured with a CHN elemental analyzer using the combustion method (Thermo Fisher Flash 2000) (Aminot and Kérouel, 2004). In addition, the BSi content was determined using the protocols adapted from Ragueneau and Tréguer (1994). Chl and Pheo were extracted in 6 ml of 90% acetone and kept in the dark at 4 °C for 12h. Samples were then centrifuged and fluorescence was measured with a Turner AU-10 fluorimeter using the equation provided in Lorenzen (1966).

The calculation of the Chl:Pheo ratio can be used to determine the freshness of the phytoplankton in the water column. According to Savoye (2001), a ratio below 1 indicates degraded organic matter, which can be considered as detrital if the value drops below 0.5, and a ratio greater than 2 describes fresh phytoplankton. At Lanvéoc in 2021, the physiological state of the phytoplankton was rather good because the ratio was always higher than 1 for the surface layer (Fig. 5), with the exception of samples from 07 June and 27 September. In bottom waters, this value occasionally fell below 2, and even below 1, i.e., on

23 March (0.82), 06 April (0.76), 20 April (1.71), 04 May (1.67) and 18 May (1.07), contemporaneously with increases in phytoplankton cells (Fig. 4). Moreover, during the main phytoplankton bloom of the year (03 June), a Chl:Pheo value below 2 was also recorded in bottom watters (1.16 on 06 June) and the minimum value was recorded a few days later (0.15), on 17 June. Between 19 July and 06 September, a time interval of low Chl:Pheo values was recorded in the bottom layer. These results indicate a degradation of phytoplankton cells that occurred after specific blooms or during summer. The reason behind

this degradation may be the natural decay of cells due to a decrease in nutrient levels, having been consumed by the blooms (Fig. 3), or due to intense grazing activities by zooplankton or heterotrophic dinoflagellates.



### 3.4 Trace element measurements

Seawater samples for trace element measurements were collected with 5 L Teflon-coated Go-Flo bottles (Teflon prevents element adsorption on the bottle walls) in surface and bottom waters. Approximately 1 L of water was filtered through 0.45 μm

MF-Millipore™ mixed cellulose ester filters (Merck). The filters were then air-dried and stored in 30 mL PTFE vials (Savillex, Minnetonka, MN, USA) until analysis of particulate trace elements. The filtrates were recovered in 15 mL polypropylene (PP) tubes and stored at (4 °C) until analysis of the dissolved trace elements. Before use, filters and PP tubes were cleaned with $HNO_3$ solutions (pH 1) made from Suprapur $HNO_3$ 65% (Merck, Darmstadt, Germany) and thoroughly rinsed with ultrapure water. PTFE vials were cleaned at 80 °C for 3 h with concentrated $HNO_3$ and $H_2O_2$ as for filter digestion (see next section) and then thoroughly rinsed with ultrapure water.

For trace element analysis of the particulate matter, filter digestions were carried out at 80 °C for 3h in closed 30 mL-PTFE screw-cap vials (Savillex, Minnetonka, MN, USA) by adding 2 mL of $HNO_3$ 65% (Merck, Darmstadt, Germany) and 500 μL of suprapur 30% hydrogen peroxide ($H_2O_2$ – Merck, Darmstadt, Germany. The elemental analyses were then conducted on diluted mixtures (2.3% $HNO_3$) with a X-series II, quadrupole inductively coupled plasma mass spectrometer (Q-ICP-MS) (Thermo Scientific) and, for Ca, P, Zn and Al specifically, with an Horiba Jobin Yvon Ultima 2 ICP - optical emission spectrometry (ICP-OES) operating at the Pôle Spectrométrie Océan (Plouzané, France). Measurements of elemental concentration within the dissolved fraction were carried out with a sector field ICP-MS (Element XR, Thermo Scientific) operating at Ifremer (with the exception of Ca and Al again, which have been measured with a ICP-OES). Since the salt matrix of seawater samples can affect chemical measurements, samples were diluted 100 times with a 2% $HNO_3$ prior to the measurement. All particulate and dissolved concentrations shown in the present study were above detection limits. The data obtained by ICP-MS or ICP-OES were also corrected for machine drift by inserting multi-element standard solutions every two samples. A certified reference materials was used: (NASS 6 seawater (National Research Council, Canada)), as well as a standard solution prepared from a multi-element solution (1 ppb - VWR chemicals - Prolabo), in order to assess the accuracy of measurements in the dissolved and the particulate phase respectively.

The network graphs show the Pearson correlations between all the trace elements measured in the particulate fraction of the surface and bottom water samples obtaining during the HIPPO monitoring (Fig. 6A and 6B, respectively). The element dynamics in the water column were clearly different between the surface layer and bottom water, i.e., the measured elements appeared strongly correlated with each other in surface water but not in the bottom water. In surface water, only Mo and Cd were not correlated with any other element, unlike Ca, Mg, V, Li, Co, Mn, Fe, Sr and Al, which formed a strongly intercorrelated group. It is worth mentioning here that a cluster appear around particulate vanadium in both water layers. Vanadium, an essential element for phytoplankton cells (Moore et al., 1996), was significantly correlated with other elements also essentials for microalgae, such as Fe, Mn or Co which are involved in several enzyme systems (Whitfield, 2001). By contrast, Ba and Sr showed patterns that differed from the other elements, and both elements were only correlated to each other in bottom water.

The temporal changes in particulate and dissolved Ba (Fig. 6C and 6D) showed interesting patterns. After the main bloom of *L. danicus* in early June 2021, the concentrations of Ba in the dissolved phase (DBa) drastically decreased, while the





concentrations of particulate barium (PBa) strongly increased both in surface and in bottom waters. Ba is not known to have any physical function for diatoms, but may be adsorbed on their frustules (Dehairs et al., 1980; Fisher et al., 1991; Sternberg et al., 2005). Therefore, findings suggest a transition of Ba from the dissolved to the particulate phase during the bloom, associated with an adsorption of Ba onto diatom frustules. An alternative explanation includes the occurrence of Acantharea
or Coccolithophoridae, known to adsorb Ba on their skeletons which are mainly composed of Sr or Ca respectively (Michaels, 1991; Bernstein et al., 1992; Langer et al., 2009). This affinity between these two elements is reflected in their significant correlation in the bottom layer only (see Figure 6B). However, no adequate method has been established to prove the presence of these organisms in 2021 in the Bay of Brest, although Coccolithophoridae cells have been observed in this region in the past, but mainly in bottom water.

### 185  3.5   Particle dynamics and composition

#### 3.5.1   Aggregation episodes

Under stress conditions, such as limitation in nutrients that had been entirely consumed during a bloom or changes in temperature, phytoplankton cells are known to excrete extracellular polymeric substances (EPS), which are precursors to transparent exopolymer particles (TEP) that favor the formation of aggregates (Alldredge et al., 1993; Passow, 2002a). These aggregates
often terminate the bloom and export the phytoplankton out of the euphotic layer to the deep oceanic layer in the open ocean, or toward the sediment (and benthic organisms) in coastal ecosystems. Particles were sized and counted using UVP6, a compact autonomous underwater imaging system (Picheral et al., 2021). This instrument has been designed for intermittent profile measurements in the water column, but here, the intention was to determine particle dynamics at the sediment-water interface as a continuous time-series. The design of the UVP6 was therefore modified to meet the needs of the present study. To minimize
turbulence, particle resuspension and natural light variation, which can alter image quality and particle detection, the objective and the light source were integrated into an opaque cylinder. This cylinder is completely closed on the bottom to avoid taking pictures of resuspended particles rising from the sediment. To ensure constant hydrodynamics within the cylinder and attenuate horizontal fluxes, a 1x1 cm mesh was added covering the top of the cylinder. This device was deployed at the seafloor on 16 March 2021. Every hour, the UVP acquired two types of data. Firstly, a picture was taken every 20 seconds for 30 minutes
to measure particle size and to calculate their concentration according to their size class (from 80.6 μmto 26 mm, divided into 25 size classes). The data are available from the ECOTAXA server (https://ecotaxa.obs-vlfr.fr) (https://ecotaxa.obs-vlfr.fr). Secondly, the UVP took full high-resolution images every 2 sec for 3 min to calculate particle trajectories and settling velocities. Due to the colossal size of these images, they are not available on SEANOE but are available upon request.

Figure 7 shows the temporal variation of small particles (daily sum of the mean concentration of particles ranging from
205 80.6 μm to 512 μm in size) and large particles (daily sum of the mean concentration of particles ranging from 512 μm to and 26 mm in size). Several aggregation periods were detected by isolating peaks in large particle concentrations whose amplitude were at least twice as large as the minimum values. Based on this method, four complete and one incomplete episodes of aggregation were identified from these time-series. The first episode took place on 30 March, the second on 26 April, the third



on 12 May, the fourth incomplete episode was recorded between 26 and 28 May (incomplete due to the removal of the UVP

to recover data and reload the battery) and, finally, the last episode on 05 June (Fig. 7). Each of these episodes was preceded by a few days-long phytoplankton bloom characterized by different species (Fig. 4B and 4C). The first aggregation episode may have resulted from the collapse of the first bloom of *C. wailesii* due to the onset of $PO_4^{3-}$ limitation (Fig. 3). After this first aggregation event, N and P were almost constantly in growth-limiting concentrations. A second period of $SiOH_4$ limitations started in early June, corresponding to the last episode of aggregation recorded during the spring and summer. In contrast,

the other aggregation events may have been triggered by other parameters. For instance, in their competition for resources, some mixotrophic dinoflagellate species such as *Gymnodinium spp.* or *Protoperidinium bipes*, which developed in late March, mid-May and early June (Fig. 4C), are known to excrete compounds toxic to other microalgae, causing an allelopathic effect and therefore cell lysis and aggregation (e.g., Legrand et al., 2003; Band-Schmidt et al., 2020).

### 3.5.2 Sediment trap data

A HYDRO-BIOS Multi Sediment Trap with 12 collecting bottles (total volume: 290 mL) was deployed during the survey to analyze the chemical composition of the particles that sedimented at the seafloor. The catchment area of this model was 153.86 $cm^2$ with a rotation frequency every three days (between 02 March and 25 May) and every four days (between 26 May and 29 July). Before deployment, each bottle was poisoned to prevent biological activity and thus degradation of the organic matter. Sedimenting material was analyzed for the trace element composition, as well as POC, PON, and biogenic silica (BSi).

Protocols used for these measurements are described in Sect. 3.3.2. and 3.4. Before filtration, the samples were shaken (to break up the aggregates) and passed through a 1 mm mesh sieve to remove big particles and avoid clogging the filters.

In total, five complete episodes of high carbon flux were recorded at Lanvéoc during 28 - 31 March (5672.57 $\mu mol.h^{-1}.m^{-2}$), 26 - 29 April (3719.93 $\mu mol.h^{-1}.m^{-2}$), 26 - 28 May (4562.25 $\mu mol.h-1.m-2$), 10 - 12 June (1887.20 $\mu mol.h^{-1}.m^{-2}$) and 19 - 21 June (5031.75 $\mu mol.h^{-1}.m^{-2}$) (Fig. 8). The average POC transport values outside these events oscillated around 700 $\mu mol.h^{-1}.m^{-2}$.

During late March, late April and late May, POC transport peaks occurred at the same time as the aggregation episodes recorded with the UVP, but the two other transport peaks did not occur at the same time as the aggregation peaks (Fig. 7). These results validate the signals obtained with the UVP and thus the deployment method of the present experiment. Furthermore, these findings highlight the role of aggregate dynamics in the transport of particulate carbon to the sediment-water interface. However, not all carbon transport events were caused by aggregation episodes, but sometimes simply by the development and the

subsequent decay of phytoplankton cells. Through the calculation of the PBa:POC ratio, Figure 8 also shows the time intervals during which an enrichment in the particulate Ba flux was recorded. PBa:POC was relatively low during the year, with a baseline value between approx. 0.01 and 0.1 $nmol.\mu mol^{-1}$. This baseline was interrupted by a large peak that occurred between 28 May and 16 June, reaching its maximum value (0.36 $nmol.\mu mol^{-1}$) on 6-9 June. During the same time, the largest diatom bloom of the year occurred (Fig. 4), as well as the subsequent aggregation episode (Fig. 7) and as the main phytoplankton

degradation event (Fig. 5). As stated in Sect. 3.4., Ba can be adsorbed onto diatom frustules, explaining the peak observed in the water column as well as the high Ba flux that occurs at cell decay after a bloom.



### 3.6 Shell data

As described in Sect. 2, a large number of shells was collected during this monitoring study. Here, only results are presented from shells collected at the end of the monitoring period, which were used to produce the growth and trace elements time-series for 2021.

#### 3.6.1 Growth rate

The *P. maximus* specimens collected during this study were all treated in the same way, whether collected directly from the sediment surface or the cage. After collection, individuals were dissected and soft tissues (gills, mantles, digestive gland and muscle) were freeze-dried for further element chemical analyses. Left (flat) valves were gently cleaned with tap water using a nylon brush and rinsed ultrasonically for 3 min with deionized water. The outer surfaces of left valves were imaged under reflected light using a Canon EOS 600 DSLR camera coupled with a Wild Heerbrugg binocular microscope equipped with a Schott VisiLED MC 1000 light source (sectoral dark field). Because the specimens were collected alive, the last growth line visible on the outer margin was formed immediately prior to the collection date. This allows to place each increment in an accurate temporal framework by counting the daily increments from the ventral margin toward the umbo, because scallop produces growth lines on a daily basis (Chauvaud et al., 1998).

The average growth rate calculated from six specimens collected from the sediment surface ranged from 69.8 to 220.6 $\mu m.day^{-1}$ with an annual average value of 171.3 $\mu m.day^{-1}$ (Fig. 9). From the onset of the growth period, growth rate gradually increased and reached a maximum value in early June. However, this increasing trend was interrupted by a slow-growth episode between 26 April and 05 May. This decline in growth rate has already been observed in previous studies of scallops from Lanvéoc, but does not occur every year (e.g., Chauvaud et al., 1998; Fröhlich et al., 2022a, b). The slow-growth phase occurred after the main bloom of *Gymnodynium sp.*, a dinoflagellate known to produce toxic blooms (Daranas et al., 2001) (Fig. 4C), and after an aggregation episode (Fig. 7). These two phenomena combined may explain the drop in shell production rate, because scallops may decrease the filtration rate during stressful conditions.

#### 3.6.2 Geochemical analyses

Geochemical analyses were performed according to protocols described in Fröhlich et al. (2022a, 2022b). The element chemical content of the shells was measured at the Max Planck Institute for Chemistry (Mainz, Germany) using a Laser Ablation (NewWave Research UP-213 Nd:YAG) - ICP-MS (Thermo Fisher Element 2) system. In total, slabe of six shells (three specimens from the sediment and three from the cage deployed 1 m above the sediment) were analyzed. Table 2 gives an exhaustive list of the elements that were measured in the present study. Laser scans were completed on every daily growth line (aka stria), by running the laser in line scan mode on the outer shell surface perpendicular to the growth direction and parallel to the growth line (Fig. 9A). Within each stria, measurements were completed using a laser spot diameter of 80 $\mu m$ at a constant speed of 5 $\mu m.sec^{-1}$. Prior to sample ablation and measurement, each sample was pre-ablated (100 $\mu m$ spot size with a speed





of 80 µm.sec$^{-1}$) to remove potential contaminants. Results are expressed as molar element-to-calcium ratios because $^{43}$Ca was used as an internal standard. Chemical data were place in accurate temporal context by means of growth pattern analysis.

Figure 10 presents the Ba/Ca data of shells collected directly on the sediment (sediment specimens) and 1 m above (cage specimens). The Ba/Ca profiles recorded in the shells are characterized by a baseline, whose values were different according to whether the shells came from the sediment (approx. 1 µmol.mol$^{-1}$ in average) or from the cage (approx. 2 µmol.mol$^{-1}$ in average). These baselines were interrupted by two major Ba/Ca peaks that occurred at approximately the same moment in all, although their amplitudes differed slightly. The first peak occurred around 07 or 08 June (sediment shells: 9.50 µmol.mol$^{-1}$, cage

shells: 6.34 µmol.mol$^{-1}$), and the second around 02 July (4.91 µmol.mol$^{-1}$ and 3.10 µmol.mol$^{-1}$, respectively). Furthermore, a third, smaller peak occurred on 30 July (2.72 µmol.mol$^{-1}$), exclusively in sediment shells. Ba content recorded in *P. maximus* shells followed more or less the same pattern as the PBa measured in bottom waters at Lanvéoc, where two major peaks were also observed at the beginning of June and July (Fig. 6D). Moreover, the relative amplitudes of these peaks were also similar, since the first peak was approximately twice as high as the second one. Nonetheless, other smaller PBa peaks occurred in

bottom waters throughout the monitoring phase, for instance, in April (Fig. 6D), and did not coincide with any Ba/Ca increase in shells (Fig. 10).

Previous studies have suggested that Ba/Ca of *P. maximus* shells can serve as a proxy for primary productivity (Barats et al., 2009) or species-specific blooms of phytoplankton (Fröhlich et al., 2022a). In the present study, the main Ba/Ca$_{shell}$ peak occurred during the main diatom peak, mainly represented by *L. danicus*, and during the main episode of Ba transport toward

the sediment (Fig. 8). The second Ba/Ca$_{shell}$ peak occurred a few days after a *Chaetoceros spp.* bloom. However, the third Ba peak recorded in sediment shells did not align with any diatom bloom. Fröhlich et al. (2022a) suggested an average time lag between diatom blooms and shell Ba peaks of 8 to 12 days, which corresponds more or less to the observations of the present study. However, the sampling frequency for phytoplankton determination was not sufficient to track the exact timing of the blooms.

## 295  4   Conclusions

In this article, only an overview of the results gathered during the HIPPO monitoring that was conducted at Lanvéoc during 2021 are presented. The dataset helps to better understand the links between phytoplankton dynamics, water column chemistry and the incorporation of trace elements into the shells of *P. maximus*. However, the dataset also contains information useful for other topics of interest. Table 1 and Table 2 compile all variables that have been made available for other scientists on the SEA-

NOE platform (https://www.seanoe.org/data/00808/92043/ - Siebert et al. (2023)). Moreover, the hypotheses and assumptions given in this paper as well as other topics that have not been mentioned will be in the focus of several articles that are currently in preparation.

*Data availability.*  https://www.seanoe.org/data/00808/92043/ - (Siebert et al., 2023)



*Author contributions.* The HIPPO monitoring has been initially designed by Julien Thébault, Bernd R. Schöne, the leader of the project.
Valentin Siebert, Brivaëla Moriceau and Julien Thébault performed the preparation of the environmental survey, with the help of several authors of this publication. During the monitoring, Erwan Amice, Thierry Lebec, Isabelle Bihannic, Emilie Grossteffan, Julien Thébault and Valentin Siebert dove at Lavéoc to collect seawater samples as well as shell materials. For several parameters, filtrations prior to measurements were conducted by Valentin Siebert or Gaspard Delebecq. POC and PON analyses were performed by Jeremy Devesa. Nutrients were measured by Manon Le Goff, with the exception of ammonium, which were measured by Emilie Grossteffan. Finally, Chlorohyll and Pheophytin were measured by Isabelle Bihannic and biogenic silicate by Morgane Gallinari and Kevin Bihannic. Aude Leynaert is in charge of the Lanvéoc observatory, where several environmental parameters are monitored on a bi-monthly basis since a few years. In the framework of this observatory, Gaspard Delebecq performed the phytoplankton identification and countings. Beatriz Beker calculated the specific biovolume of phytoplankton cells. Trace element measurements were performed by Valentin Siebert accompagnied by Matthieu Waeles for the sample preparation as well as Marie-Laure Rouget, Yoan Germain and Céline Liorzou for ICP-MS and ICP-OES analysis sessions. Lukas Fröhlich prepared the shells for LA-ICP-MS measurements, performed the measurements with Klaus-Peter Jochum, and processed the data. Lukas Fröhlich also acquired the daily growth rate data on the *P. maximus* specimens. The deployment of the UVP6 was made possible thanks to the work of Marc Picheral, who designed the instrument and kindly lent it for this monitoring. He also helped to process the data as well as Claudie Marec. Peggy Rimmelin-Maury prepared and set up the Sambat/NKE probes, discharged and processed the data. Finally, Valentin Siebert gathered all the data that were obtained for all parameters. Valentin Siebert led the writing of the manuscript. All authors contributed critically to the drafts and gave final approval for publications

*Competing interests.* The authors declar no competing interests

*Acknowledgements.* We warmly thank Brigitte Stoll from the Max Planck Institute (Mainz) for her help on the LA-ICP-MS analyses. We are also very grateful to the crew of the Albert Lucas for their involvement in this environmental monitoring and their support during the cruises. We also thank the Plateau d'analyse chimique des paramètres de base de l'environnement marin (PACHIDERM) as well as the Pôle Spectrométrie Océan (PSO) for the measurements of the various chemical parameters.

The HIPPO monitoring was funded by the French National Research Agency (Agence Nationale de la Recherche - ANR) and a German Research Foundation (the Deutsche Forschungsgemeinschaft - DFG) with the support of the Région Bretagne. This survey has been performed within the framework of the French-German collaborative project HIPPO (HIgh-resolution Primary Production multiprOxy archives).



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

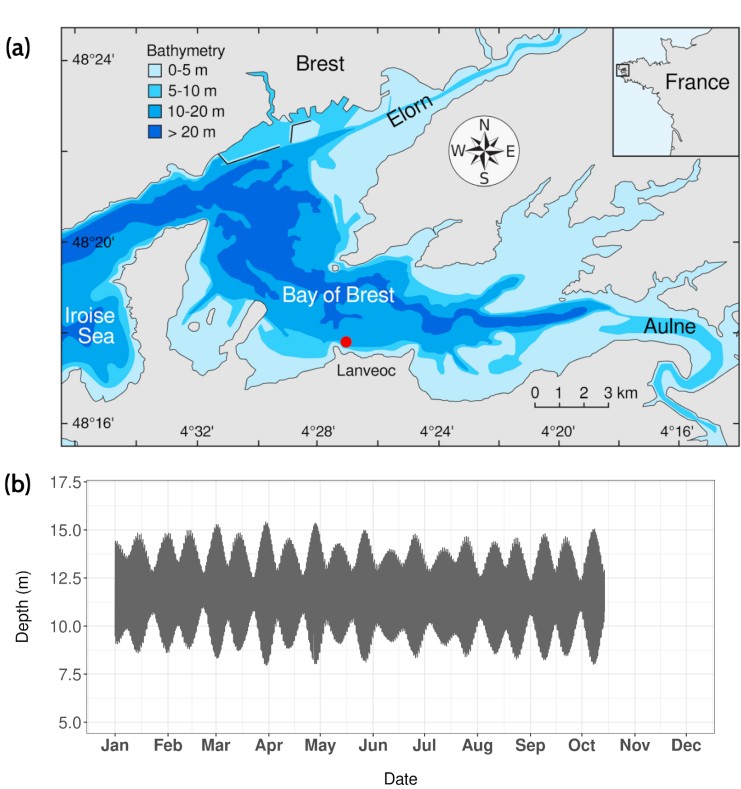

**Figure 1.** (a) Sampling location of the HIPPO monitoring area: Lanvéoc (red dot) (adapted from Thébault et al., 2022). (b) Hourly depth variations recorded with NKE/Sambat probes deployed at the seafloor at Lanvéoc in 2021.



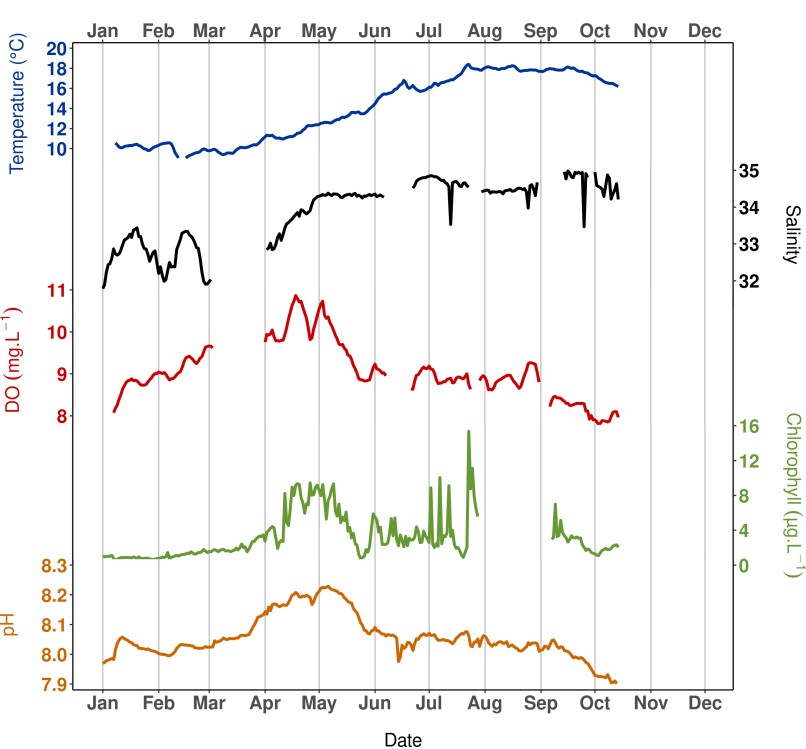

**Figure 2.** Temperature, salinity, dissolved oxygen (DO), chlorophyll concentration computed from fluorescence and pH recorded with a NKE/Sambat probe deployed at the seafloor at Lanvéoc during 2021.



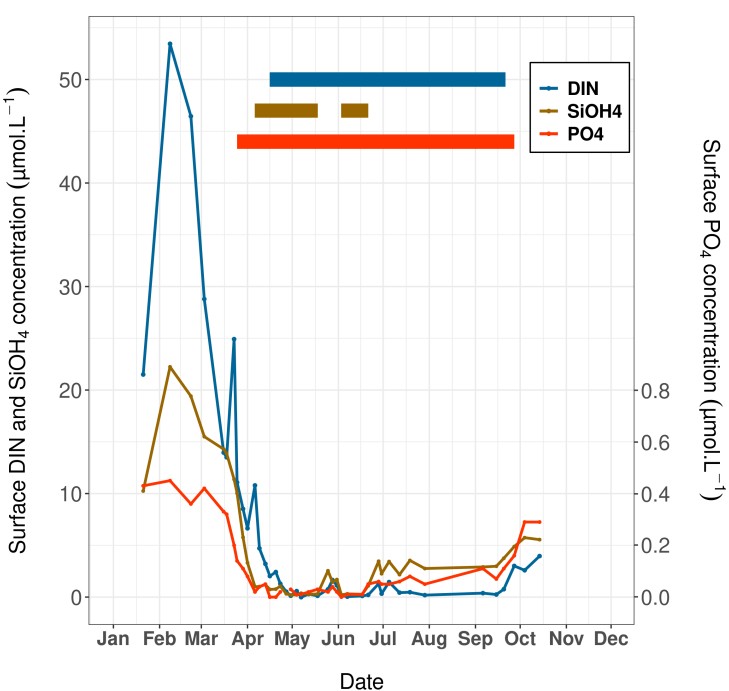

**Figure 3.** Surface nutrient concentrations (DIN, silicates and phosphates) over time measured at Lanvéoc during 2021; their respective periods of limitation (thick solid lines) are indicated.







**Figure 4.** (a) Phytoplankton dynamics recorded at Lanvéoc in 2021. Relative presence of the main (b) diatoms and (c) dinoflagellate species expressed as cell biovolumes.

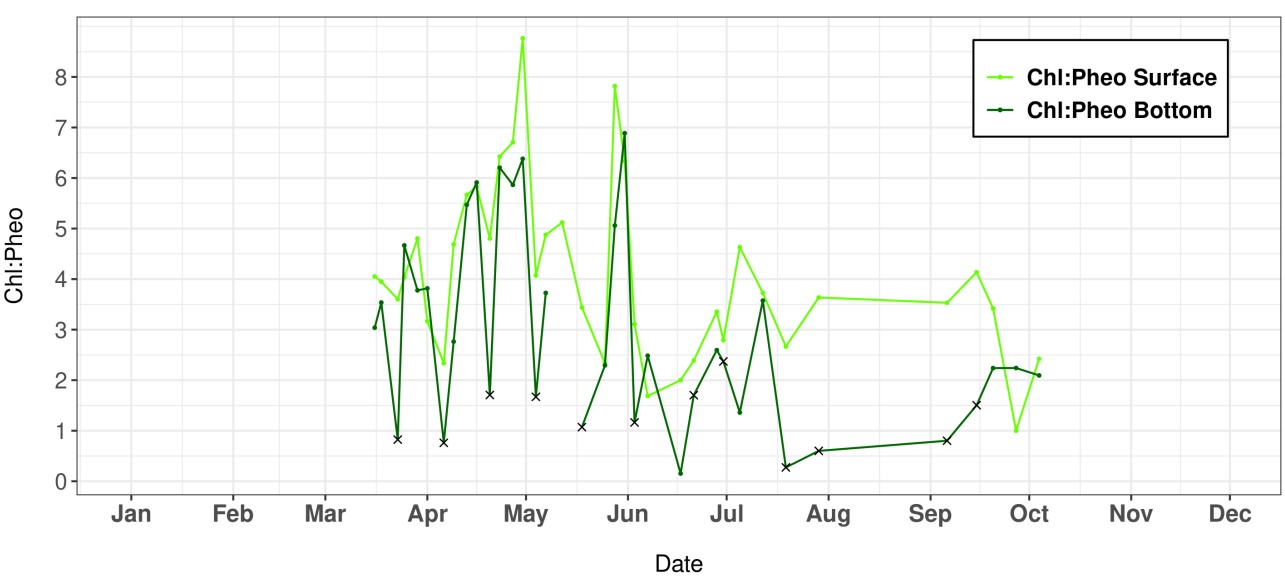

**Figure 5.** Variations in the chlorophyll:pheophytin ratio (Chl:Pheo) measured in surface and bottom layers of the water column at Lanvéoc during 2021. Crosses indicate seawater sampled with a bottom syringe.





**Figure 6.** Network graphs showing Pearson correlations between each measured trace element of the particulate fraction in (a) surface and (b) bottom waters. Only significant correlations with a coefficient greater than 0.62 are shown on these graphs. (c) Dissolved and (d) particulate barium (DBa and PBa, respectively) recorded at Lanvéoc during 2021 in surface and bottom waters.

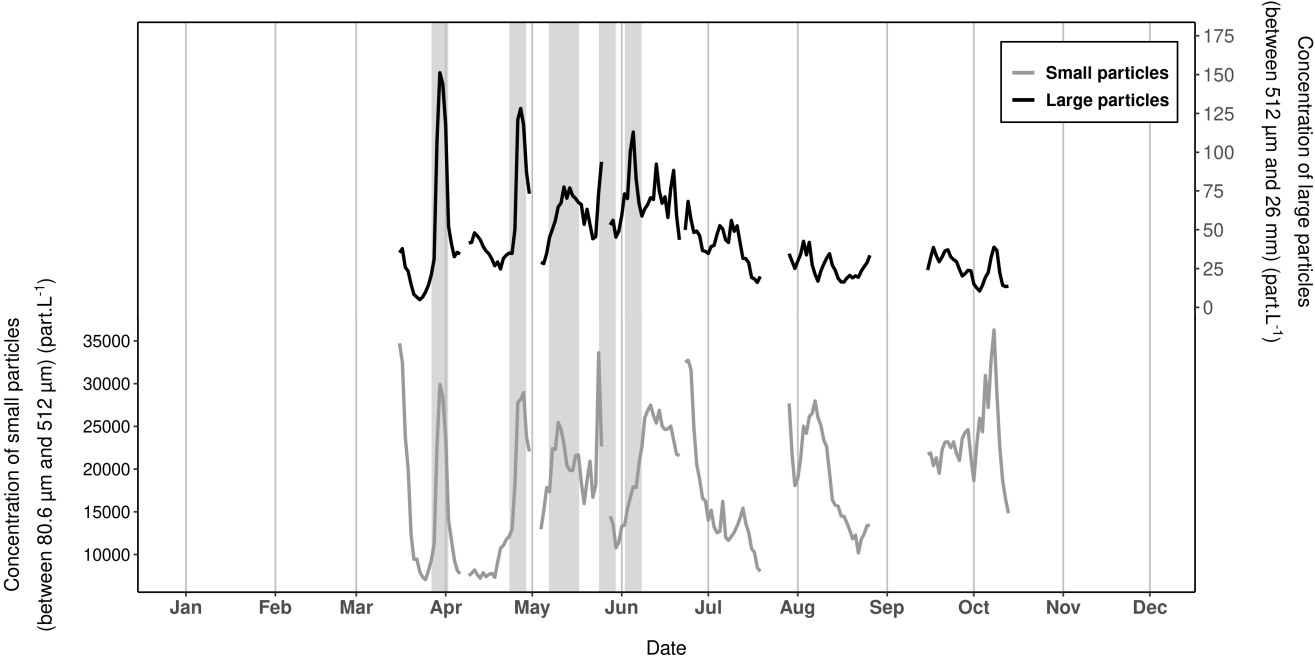

**Figure 7.** Variations in the daily mean particle concentrations recorded with the UVP6 imaging sensor at Lanvéoc during 2021. The dark lines represent the concentration of large particles between 512 µm and 4.10 mm (equivalent spherical diameter, ESD) and the gray lines show the concentration of smaller particles, ranging from 50.8 to 512 µm (ESD). Grey-shaded areas denote aggregation events.

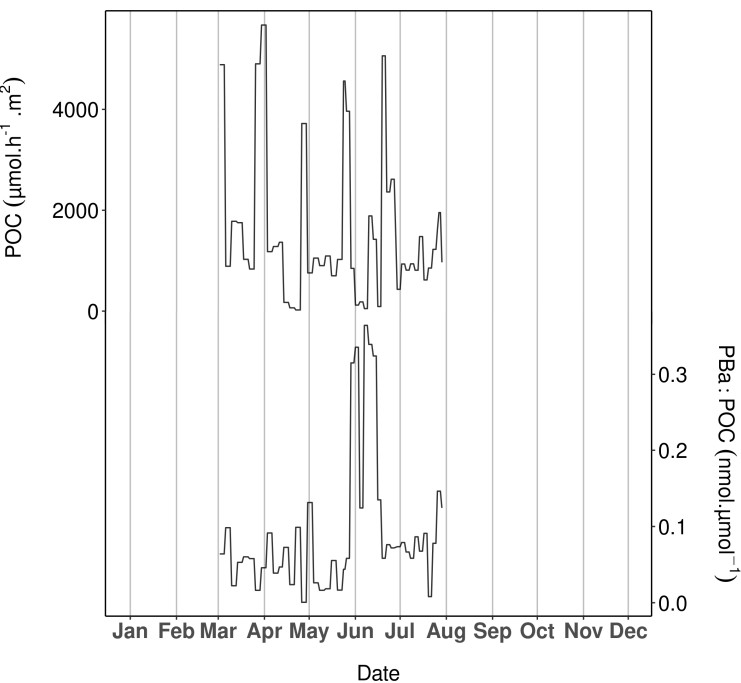

**Figure 8.** Particulate organic carbon (POC) (upper curve) and the particulate barium-to-POC ratio (PBa:POC) (lower curve) measured in the sediment trap samples at Lanvéoc in 2021.

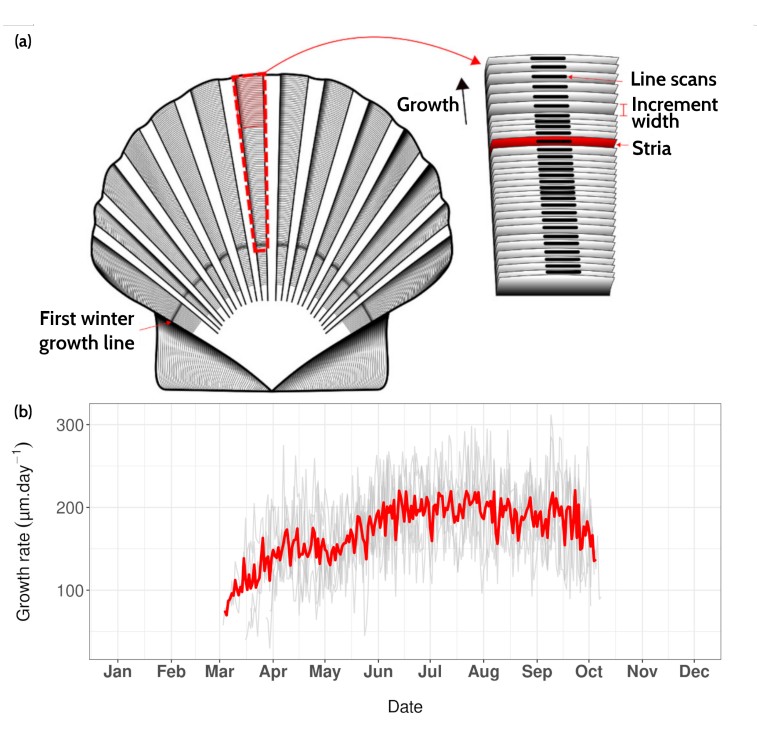

**Figure 9.** (a) Surface of *Pecten maximus* shell (left valve), showing growth patterns (increments and lines, aka 'striae') as well as the sampling strategy for trace element measurements via LA-ICP-MS ("line scans") (from Fröhlich et al. (2022a)). (b) Average daily growth rate (red curve) calculated from 6 specimens (grey curves) that were collected alive at Lanvéoc from the sediment surface.

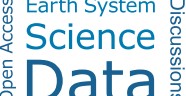

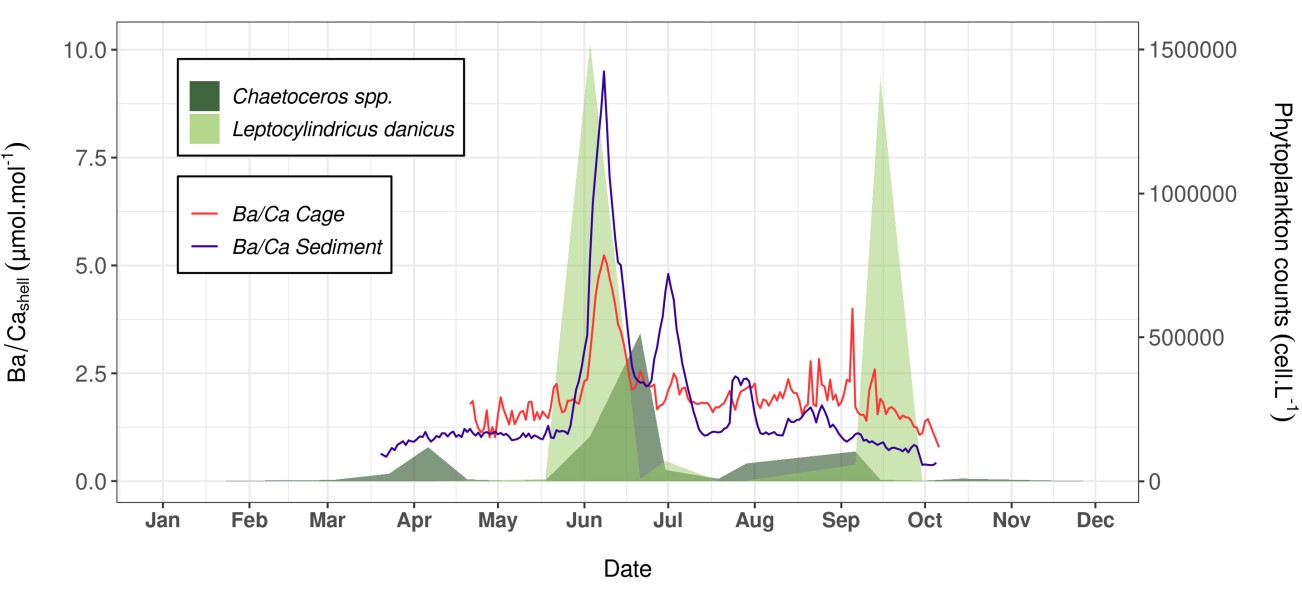

**Figure 10.** Average Ba/Ca signals measured in shell of *P. maximus* that were collected from the sediment surface (n=3) and 1 m above the substrate (n =3). The abundances of *Chaetoceros spp.* (dark green areas) and *L. danicus* (light green areas) are also shown





**Table 1.** List of environmental parameters measured at Lanvéoc in 2021 during the HIPPO monitoring.

| Date of first and last acquisition | Parameter | Sampling method | Water column layers | Frequency | Principals investigators | Processed |
|---|---|---|---|---|---|---|
| 16.03.2021 - 14.10.2021 | POC | Niskin bottle | Surface | Every 3 or 4 days | Jérémy Devesa | Available |
| 16.03.2021 - 14.10.2021 | POC | Niskin bottle | Bottom | Every 3 or 4 days | Jérémy Devesa | Available |
| 02.03.2021 - 29.07.2021 | POC | Sediment trap | Bottom | Every 3 or 4 days | Jérémy Devesa | Available |
| 07.01.2021 - 13.12.2021 | POC | Bottom Syringe | Bottom | Every 2 weeks | Jérémy Devesa | Available |
| 16.03.2021 - 14.10.2021 | PON | Niskin bottle | Surface | Every 3 or 4 days | Jérémy Devesa | Available |
| 16.03.2021 - 14.10.2021 | PON | Niskin bottle | Bottom | Every 3 or 4 days | Jérémy Devesa | Available |
| 02.03.2021 - 29.07.2021 | PON | Sediment trap | Bottom | Every 3 or 4 days | Jérémy Devesa | Available |
| 07.01.2021 – 13.12.2021 | PON | Bottom Syringe | Bottom | Every 2 weeks | Jérémy Devesa | Available |
| 16.03.2021 - 14.10.2021 | BSi | Niskin bottle | Surface | Every 3 or 4 days | Morgane Gallinari | Available |
| 16.03.2021 - 14.10.2021 | BSi | Niskin bottle | Bottom | Every 3 or 4 days | Morgane Gallinari | Available |
| 02.03.2021 - 29.07.2021 | BSi | Sediment trap | Bottom | Every 3 or 4 days | Morgane Gallinari | Available |
| 16.03.2021 - 14.10.2021 | $NH_4^+$ | Niskin bottle | Surface | Every 3 or 4 days | Émilie Grossteffan | Available |
| 16.03.2021 - 14.10.2021 | $NO_3$ | Niskin bottle | Surface | Every 3 or 4 days | Manon Le Goff | Available |
| 16.03.2021 - 14.10.2021 | $NO_2$ | Niskin bottle | Surface | Every 3 or 4 days | Manon Le Goff | Available |
| 16.03.2021 - 14.10.2021 | $SiOH_4$ | Niskin bottle | Surface | Every 3 or 4 days | Manon Le Goff | Available |
| 16.03.2021 - 14.10.2021 | $PO_4^{3-}$ | Niskin bottle | Surface | Every 3 or 4 days | Manon Le Goff | Available |
| 16.03.2021 - 14.10.2021 | Chlorophyll a | Niskin bottle | Surface | Every 3 or 4 days | Isabelle Bihannic | Available |
| 16.03.2021 - 14.10.2021 | Chlorophyll a | Niskin bottle | Bottom | Every 3 or 4 days | Isabelle Bihannic | Available |
| 07.01.2021 - 13.12.2021 | Chlorophyll a | Bottom Syringe | Bottom | Every 2 weeks | Isabelle Bihannic | Available |
| 16.03.2021 - 14.10.2021 | Pheophytin | Niskin bottle | Surface | Every 3 or 4 days | Isabelle Bihannic | Available |
| 16.03.2021 - 14.10.2021 | Pheophytin | Niskin bottle | Bottom | Every 3 or 4 days | Isabelle Bihannic | Available |
| 07.01.2021 - 13.12.2021 | Pheophytin | Bottom Syringe | Bottom | Every 2 weeks | Isabelle Bihannic | Available |
| 01.01.2021 - 31.12.2021 | Phytoplankton (taxonomy and counting) | Niskin bottle | Surface | Every 2 weeks | Gaspard Delebecq | Only available on request |





| Date | Parameter | Instrument | Position | Frequency | Responsible | Status |
|---|---|---|---|---|---|---|
| 01.01.2021 - 31.12.2021 | Temperature | Sambat probes | Bottom | Every 20 minutes | Peggy Rimmelin-Maury | Available |
| 01.01.2021 - 31.12.2021 | Salinity | Sambat probes | Bottom | Every 20 minutes | Peggy Rimmelin-Maury | Available |
| 01.01.2021 - 31.12.2021 | Oxygen | Sambat probes | Bottom | Every 20 minutes | Peggy Rimmelin-Maury | Available |
| 01.01.2021 - 31.12.2021 | pH | Sambat probes | Bottom | Every 20 minutes | Peggy Rimmelin-Maury | Available |
| 16.03.2021 - 14.10.2021 | Fluorescence | Sambat probes | Bottom | Every 20 minutes | Peggy Rimmelin-Maury | Available |
| 01.01.2021 - 31.12.2021 | Depth | Sambat probes | | Every 20 minutes | Peggy Rimmelin-Maury | Available |
| 16.03.2021 - 14.10.2021 | Particles concentration | UVP6 | Bottom | Every 30 minutes | Marc Picheral | Available |
| 16.03.2021 - 14.10.2021 | Particles size distribution | UVP6 | Bottom | Every 30 minutes | Marc Picheral | Available |
| 16.03.2021 - 14.10.2021 | Particles identification and characterisation | UVP6 | Bottom | Every 30 minutes | Marc Picheral | Available |
| 16.03.2021 - 14.10.2021 | Particulate $^7$Lithium | GoFlo Bottle | Surface | Every 3 or 4 days | Valentin Siebert - Marie Laure Rouget | Available |
| 16.03.2021 - 14.10.2021 | Particulate $^7$Lithium | GoFlo Bottle | Bottom | Every 3 or 4 days | Valentin Siebert - Marie Laure Rouget | Available |
| 02.03.2021 - 29.07.2021 | Particulate $^7$Lithium | Sediment trap | | Every 3 or 4 days | Valentin Siebert - Marie Laure Rouget | Available |
| 16.03.2021 - 14.10.2021 | Particulate $^{25}$Magnesium | GoFlo Bottle | Surface | Every 3 or 4 days | Valentin Siebert - Marie Laure Rouget | Available |
| 16.03.2021 - 14.10.2021 | Particulate $^{25}$Magnesium | GoFlo Bottle | Bottom | Every 3 or 4 days | Valentin Siebert - Marie Laure Rouget | Available |
| 02.03.2021 - 29.07.2021 | Particulate $^{25}$Magnesium | Sediment trap | | Every 3 or 4 days | Valentin Siebert - Marie Laure Rouget | Available |
| 16.03.2021 - 14.10.2021 | Particulate $^{31}$Phosphorus | GoFlo Bottle | Surface | Every 3 or 4 days | Valentin Siebert - Marie Laure Rouget | Available |
| 16.03.2021 - 14.10.2021 | Particulate $^{31}$Phosphorus | GoFlo Bottle | Bottom | Every 3 or 4 days | Valentin Siebert - Marie Laure Rouget | Available |
| 02.03.2021 - 29.07.2021 | Particulate $^{31}$Phosphorus | Sediment trap | | Every 3 or 4 days | Valentin Siebert - Marie Laure Rouget | Available |
| 16.03.2021 - 14.10.2021 | Particulate $^{51}$Vanadium | GoFlo Bottle | Surface | Every 3 or 4 days | Valentin Siebert - Marie Laure Rouget | Available |
| 16.03.2021 - 14.10.2021 | Particulate $^{51}$Vanadium | GoFlo Bottle | Bottom | Every 3 or 4 days | Valentin Siebert - Marie Laure Rouget | Available |
| 02.03.2021 - 29.07.2021 | Particulate $^{51}$Vanadium | Sediment trap | | Every 3 or 4 days | Valentin Siebert - Marie Laure Rouget | Available |
| 16.03.2021 - 14.10.2021 | Particulate $^{52}$Chromium | GoFlo Bottle | Surface | Every 3 or 4 days | Valentin Siebert - Marie Laure Rouget | Available |
| 16.03.2021 - 14.10.2021 | Particulate $^{52}$Chromium | GoFlo Bottle | Bottom | Every 3 or 4 days | Valentin Siebert - Marie Laure Rouget | Available |
| 02.03.2021 - 29.07.2021 | Particulate $^{52}$Chromium | Sediment trap | | Every 3 or 4 days | Valentin Siebert - Marie Laure Rouget | Available |
| 16.03.2021 - 14.10.2021 | Particulate $^{55}$Manganese | GoFlo Bottle | Surface | Every 3 or 4 days | Valentin Siebert - Marie Laure Rouget | Available |





| | | | | | | |
|---|---|---|---|---|---|---|
| 16.03.2021 - 14.10.2021 | Particulate $^{55}$Manganese | GoFlo Bottle | Bottom | Every 3 or 4 days | Valentin Siebert - Marie Laure Rouget | Available |
| 02.03.2021 - 29.07.2021 | Particulate $^{55}$Manganese | Sediment trap | | Every 3 or 4 days | Valentin Siebert - Marie Laure Rouget | Available |
| 16.03.2021 - 14.10.2021 | Particulate $^{56}$Iron | GoFlo Bottle | Surface | Every 3 or 4 days | Valentin Siebert - Marie Laure Rouget | Available |
| 16.03.2021 - 14.10.2021 | Particulate $^{56}$Iron | GoFlo Bottle | Bottom | Every 3 or 4 days | Valentin Siebert - Marie Laure Rouget | Available |
| 02.03.2021 - 29.07.2021 | Particulate $^{56}$Iron | Sediment trap | | Every 3 or 4 days | Valentin Siebert - Marie Laure Rouget | Available |
| 16.03.2021 - 14.10.2021 | Particulate $^{59}$Cobalt | GoFlo Bottle | Surface | Every 3 or 4 days | Valentin Siebert - Marie Laure Rouget | Available |
| 16.03.2021 - 14.10.2021 | Particulate $^{59}$Cobalt | GoFlo Bottle | Bottom | Every 3 or 4 days | Valentin Siebert - Marie Laure Rouget | Available |
| 02.03.2021 - 29.07.2021 | Particulate $^{59}$Cobalt | Sediment trap | | Every 3 or 4 days | Valentin Siebert - Marie Laure Rouget | Available |
| 16.03.2021 - 14.10.2021 | Particulate $^{63}$Copper | GoFlo Bottle | Surface | Every 3 or 4 days | Valentin Siebert - Marie Laure Rouget | Available |
| 16.03.2021 - 14.10.2021 | Particulate $^{63}$Copper | GoFlo Bottle | Bottom | Every 3 or 4 days | Valentin Siebert - Marie Laure Rouget | Available |
| 02.03.2021 - 29.07.2021 | Particulate $^{63}$Copper | Sediment trap | | Every 3 or 4 days | Valentin Siebert - Marie Laure Rouget | Available |
| 16.03.2021 - 14.10.2021 | Particulate $^{66}$Zinc | GoFlo Bottle | Surface | Every 3 or 4 days | Valentin Siebert - Marie Laure Rouget | Available |
| 16.03.2021 - 14.10.2021 | Particulate $^{66}$Zinc | GoFlo Bottle | Bottom | Every 3 or 4 days | Valentin Siebert - Marie Laure Rouget | Available |
| 02.03.2021 - 29.07.2021 | Particulate $^{66}$Zinc | Sediment trap | | Every 3 or 4 days | Valentin Siebert - Marie Laure Rouget | Available |
| 16.03.2021 - 14.10.2021 | Particulate $^{75}$Arsenic | GoFlo Bottle | Surface | Every 3 or 4 days | Valentin Siebert - Marie Laure Rouget | Available |
| 16.03.2021 - 14.10.2021 | Particulate $^{75}$Arsenic | GoFlo Bottle | Bottom | Every 3 or 4 days | Valentin Siebert - Marie Laure Rouget | Available |
| 02.03.2021 - 29.07.2021 | Particulate $^{75}$Arsenic | Sediment trap | | Every 3 or 4 days | Valentin Siebert - Marie Laure Rouget | Available |
| 16.03.2021 - 14.10.2021 | Particulate $^{88}$Strontium | GoFlo Bottle | Surface | Every 3 or 4 days | Valentin Siebert - Marie Laure Rouget | Available |
| 16.03.2021 - 14.10.2021 | Particulate $^{88}$Strontium | GoFlo Bottle | Bottom | Every 3 or 4 days | Valentin Siebert - Marie Laure Rouget | Available |
| 02.03.2021 - 29.07.2021 | Particulate $^{88}$Strontium | Sediment trap | | Every 3 or 4 days | Valentin Siebert - Marie Laure Rouget | Available |
| 16.03.2021 - 14.10.2021 | Particulate $^{95}$Molybdenum | GoFlo Bottle | Surface | Every 3 or 4 days | Valentin Siebert - Marie Laure Rouget | Available |
| 16.03.2021 - 14.10.2021 | Particulate $^{95}$Molybdenum | GoFlo Bottle | Bottom | Every 3 or 4 days | Valentin Siebert - Marie Laure Rouget | Available |
| 02.03.2021 - 29.07.2021 | Particulate $^{95}$Molybdenum | Sediment trap | | Every 3 or 4 days | Valentin Siebert - Marie Laure Rouget | Available |
| 16.03.2021 - 14.10.2021 | Particulate $^{137}$Barium | GoFlo Bottle | Surface | Every 3 or 4 days | Valentin Siebert - Marie Laure Rouget | Available |
| 16.03.2021 - 14.10.2021 | Particulate $^{137}$Barium | GoFlo Bottle | Bottom | Every 3 or 4 days | Valentin Siebert - Marie Laure Rouget | Available |



| | | | | | | |
|---|---|---|---|---|---|---|
| 02.03.2021 - 29.07.2021 | Particulate $^{137}$Barium | Sediment trap | | Every 3 or 4 days | Valentin Siebert - Marie Laure Rouget | Available |
| 16.03.2021 - 14.10.2021 | Particulate $^{138}$Barium | GoFlo Bottle | Surface | Every 3 or 4 days | Valentin Siebert - Marie Laure Rouget | Available |
| 16.03.2021 - 14.10.2021 | Particulate $^{138}$Barium | GoFlo Bottle | Bottom | Every 3 or 4 days | Valentin Siebert - Marie Laure Rouget | Available |
| 02.03.2021 - 29.07.2021 | Particulate $^{138}$Barium | Sediment trap | | Every 3 or 4 days | Valentin Siebert - Marie Laure Rouget | Available |
| 16.03.2021 - 14.10.2021 | Particulate $^{206}$Lead | GoFlo Bottle | Surface | Every 3 or 4 days | Valentin Siebert - Marie Laure Rouget | Available |
| 16.03.2021 - 14.10.2021 | Particulate $^{206}$Lead | GoFlo Bottle | Bottom | Every 3 or 4 days | Valentin Siebert - Marie Laure Rouget | Available |
| 02.03.2021 - 29.07.2021 | Particulate $^{206}$Lead | Sediment trap | | Every 3 or 4 days | Valentin Siebert - Marie Laure Rouget | Available |
| 16.03.2021 - 14.10.2021 | Particulate $^{207}$Lead | GoFlo Bottle | Surface | Every 3 or 4 days | Valentin Siebert - Marie Laure Rouget | Available |
| 16.03.2021 - 14.10.2021 | Particulate $^{207}$Lead | GoFlo Bottle | Bottom | Every 3 or 4 days | Valentin Siebert - Marie Laure Rouget | Available |
| 02.03.2021 - 29.07.2021 | Particulate $^{207}$Lead | Sediment trap | | Every 3 or 4 days | Valentin Siebert - Marie Laure Rouget | Available |
| 16.03.2021 - 14.10.2021 | Particulate $^{208}$Lead | GoFlo Bottle | Surface | Every 3 or 4 days | Valentin Siebert - Marie Laure Rouget | Available |
| 16.03.2021 - 14.10.2021 | Particulate $^{208}$Lead | GoFlo Bottle | Bottom | Every 3 or 4 days | Valentin Siebert - Marie Laure Rouget | Available |
| 02.03.2021 - 29.07.2021 | Particulate $^{208}$Lead | Sediment trap | | Every 3 or 4 days | Valentin Siebert - Marie Laure Rouget | Available |
| 16.03.2021 - 14.10.2021 | Dissolved $^{7}$Lithium | GoFlo Bottle | Surface | Every 3 or 4 days | Valentin Siebert - Yoan Germain | Available |
| 16.03.2021 - 14.10.2021 | Dissolved $^{7}$Lithium | GoFlo Bottle | Bottom | Every 3 or 4 days | Valentin Siebert - Yoan Germain | Available |
| 16.03.2021 - 14.10.2021 | Dissolved $^{25}$Magnesium | GoFlo Bottle | Surface | Every 3 or 4 days | Valentin Siebert - Yoan Germain | Available |
| 16.03.2021 - 14.10.2021 | Dissolved $^{25}$Magnesium | GoFlo Bottle | Bottom | Every 3 or 4 days | Valentin Siebert - Yoan Germain | Available |
| 16.03.2021 - 14.10.2021 | Dissolved $^{31}$Phosphorus | GoFlo Bottle | Surface | Every 3 or 4 days | Valentin Siebert - Yoan Germain | Available |
| 16.03.2021 - 14.10.2021 | Dissolved $^{31}$Phosphorus | GoFlo Bottle | Bottom | Every 3 or 4 days | Valentin Siebert - Yoan Germain | Available |
| 16.03.2021 - 14.10.2021 | Dissolved $^{51}$Vanadium | GoFlo Bottle | Surface | Every 3 or 4 days | Valentin Siebert - Yoan Germain | Available |
| 16.03.2021 - 14.10.2021 | Dissolved $^{51}$Vanadium | GoFlo Bottle | Bottom | Every 3 or 4 days | Valentin Siebert - Yoan Germain | Available |
| 16.03.2021 - 14.10.2021 | Dissolved $^{52}$Chromium | GoFlo Bottle | Surface | Every 3 or 4 days | Valentin Siebert - Yoan Germain | Available |
| 16.03.2021 - 14.10.2021 | Dissolved $^{52}$Chromium | GoFlo Bottle | Bottom | Every 3 or 4 days | Valentin Siebert - Yoan Germain | Available |
| 16.03.2021 - 14.10.2021 | Dissolved $^{55}$Manganese | GoFlo Bottle | Surface | Every 3 or 4 days | Valentin Siebert - Yoan Germain | Available |
| 16.03.2021 - 14.10.2021 | Dissolved $^{55}$Manganese | GoFlo Bottle | Bottom | Every 3 or 4 days | Valentin Siebert - Yoan Germain | Available |



| | | | | | | |
|---|---|---|---|---|---|---|
| 16.03.2021 - 14.10.2021 | Dissolved $^{56}$Iron | GoFlo Bottle | Surface | Every 3 or 4 days | Valentin Siebert - Yoan Germain | Available |
| 16.03.2021 - 14.10.2021 | Dissolved $^{56}$Iron | GoFlo Bottle | Bottom | Every 3 or 4 days | Valentin Siebert - Yoan Germain | Available |
| 16.03.2021 - 14.10.2021 | Dissolved $^{59}$Cobalt | GoFlo Bottle | Surface | Every 3 or 4 days | Valentin Siebert - Yoan Germain | Available |
| 16.03.2021 - 14.10.2021 | Dissolved $^{59}$Cobalt | GoFlo Bottle | Bottom | Every 3 or 4 days | Valentin Siebert - Yoan Germain | Available |
| 16.03.2021 - 14.10.2021 | Dissolved $^{61}$Nickel | GoFlo Bottle | Surface | Every 3 or 4 days | Valentin Siebert - Yoan Germain | Available |
| 16.03.2021 - 14.10.2021 | Dissolved $^{61}$Nickel | GoFlo Bottle | Bottom | Every 3 or 4 days | Valentin Siebert - Yoan Germain | Available |
| 16.03.2021 - 14.10.2021 | Dissolved $^{63}$Copper | GoFlo Bottle | Surface | Every 3 or 4 days | Valentin Siebert - Yoan Germain | Available |
| 16.03.2021 - 14.10.2021 | Dissolved $^{63}$Copper | GoFlo Bottle | Bottom | Every 3 or 4 days | Valentin Siebert - Yoan Germain | Available |
| 16.03.2021 - 14.10.2021 | Dissolved $^{66}$Zinc | GoFlo Bottle | Surface | Every 3 or 4 days | Valentin Siebert - Yoan Germain | Available |
| 16.03.2021 - 14.10.2021 | Dissolved $^{66}$Zinc | GoFlo Bottle | Bottom | Every 3 or 4 days | Valentin Siebert - Yoan Germain | Available |
| 16.03.2021 - 14.10.2021 | Dissolved $^{75}$Arsenic | GoFlo Bottle | Surface | Every 3 or 4 days | Valentin Siebert - Yoan Germain | Available |
| 16.03.2021 - 14.10.2021 | Dissolved $^{75}$Arsenic | GoFlo Bottle | Bottom | Every 3 or 4 days | Valentin Siebert - Yoan Germain | Available |
| 16.03.2021 - 14.10.2021 | Dissolved $^{88}$Strontium | GoFlo Bottle | Surface | Every 3 or 4 days | Valentin Siebert - Yoan Germain | Available |
| 16.03.2021 - 14.10.2021 | Dissolved $^{88}$Strontium | GoFlo Bottle | Bottom | Every 3 or 4 days | Valentin Siebert - Yoan Germain | Available |
| 16.03.2021 - 14.10.2021 | Dissolved $^{95}$Molybdenum | GoFlo Bottle | Surface | Every 3 or 4 days | Valentin Siebert - Yoan Germain | Available |
| 16.03.2021 - 14.10.2021 | Dissolved $^{95}$Molybdenum | GoFlo Bottle | Bottom | Every 3 or 4 days | Valentin Siebert - Yoan Germain | Available |
| 16.03.2021 - 14.10.2021 | Dissolved $^{111}$Cadmium | GoFlo Bottle | Surface | Every 3 or 4 days | Valentin Siebert - Yoan Germain | Available |
| 16.03.2021 - 14.10.2021 | Dissolved $^{111}$Cadmium | GoFlo Bottle | Bottom | Every 3 or 4 days | Valentin Siebert - Yoan Germain | Available |
| 16.03.2021 - 14.10.2021 | Dissolved $^{137}$Barium | GoFlo Bottle | Surface | Every 3 or 4 days | Valentin Siebert - Yoan Germain | Available |
| 16.03.2021 - 14.10.2021 | Dissolved $^{137}$Barium | GoFlo Bottle | Bottom | Every 3 or 4 days | Valentin Siebert - Yoan Germain | Available |
| 16.03.2021 - 14.10.2021 | Dissolved $^{138}$Barium | GoFlo Bottle | Surface | Every 3 or 4 days | Valentin Siebert - Yoan Germain | Available |
| 16.03.2021 - 14.10.2021 | Dissolved $^{138}$Barium | GoFlo Bottle | Bottom | Every 3 or 4 days | Valentin Siebert - Yoan Germain | Available |
| 16.03.2021 - 14.10.2021 | Dissolved $^{206}$Lead | GoFlo Bottle | Surface | Every 3 or 4 days | Valentin Siebert - Yoan Germain | Available |
| 16.03.2021 - 14.10.2021 | Dissolved $^{206}$Lead | GoFlo Bottle | Bottom | Every 3 or 4 days | Valentin Siebert - Yoan Germain | Available |
| 16.03.2021 - 14.10.2021 | Dissolved $^{207}$Lead | GoFlo Bottle | Surface | Every 3 or 4 days | Valentin Siebert - Yoan Germain | Available |



| 16.03.2021 - 14.10.2021 | Dissolved [207]Lead | GoFlo Bottle | Bottom | Every 3 or 4 days | Valentin Siebert - Yoan Germain | Available |
|---|---|---|---|---|---|---|
| 16.03.2021 - 14.10.2021 | Dissolved [208]Lead | GoFlo Bottle | Surface | Every 3 or 4 days | Valentin Siebert - Yoan Germain | Available |
| 16.03.2021 - 14.10.2021 | Dissolved [208]Lead | GoFlo Bottle | Bottom | Every 3 or 4 days | Valentin Siebert - Yoan Germain | Available |



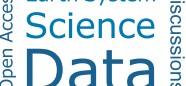

**Table 2.** List of parameters measured within the shells of *Pecten maximus* collected at Lanvéoc in 2021.

| Parameter | Sediment/Cage | Number of analysed shells | Principals investigators | Processed |
|---|---|---|---|---|
| Daily growth rate | Sediment | 6 | Lukas Fröhlich – Valentin Siebert | Available |
| Daily growth rate | Cage | 6 | Lukas Fröhlich – Valentin Siebert | Available |
| $^7$Li / $^{43}$Ca$_{shell}$ | Sediment | 3 | Lukas Fröhlich – Valentin Siebert | Available |
| $^{11}$B / $^{43}$Ca$_{shell}$ | Sediment | 3 | Lukas Fröhlich – Valentin Siebert | Available |
| $^{23}$Na / $^{43}$Ca$_{shell}$ | Sediment | 3 | Lukas Fröhlich – Valentin Siebert | Available |
| $^{25}$Mg / $^{43}$Ca$_{shell}$ | Sediment | 3 | Lukas Fröhlich – Valentin Siebert | Available |
| $^{43}$Ca / $^{43}$Ca$_{shell}$ | Sediment | 3 | Lukas Fröhlich – Valentin Siebert | Available |
| $^{51}$V / $^{43}$Ca$_{shell}$ | Sediment | 3 | Lukas Fröhlich – Valentin Siebert | Available |
| $^{55}$Mn / $^{43}$Ca$_{shell}$ | Sediment | 3 | Lukas Fröhlich – Valentin Siebert | Available |
| $^{57}$Fe / $^{43}$Ca$_{shell}$ | Sediment | 3 | Lukas Fröhlich – Valentin Siebert | Available |
| $^{88}$Sr / $^{43}$Ca$_{shell}$ | Sediment | 3 | Lukas Fröhlich – Valentin Siebert | Available |
| $^{95}$Mo / $^{43}$Ca$_{shell}$ | Sediment | 3 | Lukas Fröhlich – Valentin Siebert | Available |
| $^{97}$Mo / $^{43}$Ca$_{shell}$ | Sediment | 3 | Lukas Fröhlich – Valentin Siebert | Available |
| $^{135}$Ba / $^{43}$Ca$_{shell}$ | Sediment | 3 | Lukas Fröhlich – Valentin Siebert | Available |
| $^{137}$Ba / $^{43}$Ca$_{shell}$ | Sediment | 3 | Lukas Fröhlich – Valentin Siebert | Available |
| $^{208}$Pb / $^{43}$Ca$_{shell}$ | Sediment | 3 | Lukas Fröhlich – Valentin Siebert | Available |
| $^{238}$U / $^{43}$Ca$_{shell}$ | Sediment | 3 | Lukas Fröhlich – Valentin Siebert | Available |
| $^7$Li / $^{43}$Ca$_{shell}$ | Cage | 3 | Lukas Fröhlich – Valentin Siebert | Available |
| $^{11}$B / $^{43}$Ca$_{shell}$ | Cage | 3 | Lukas Fröhlich – Valentin Siebert | Available |
| $^{23}$Na / $^{43}$Ca$_{shell}$ | Cage | 3 | Lukas Fröhlich – Valentin Siebert | Available |
| $^{25}$Mg / $^{43}$Ca$_{shell}$ | Cage | 3 | Lukas Fröhlich – Valentin Siebert | Available |
| $^{43}$Ca / $^{43}$Ca$_{shell}$ | Cage | 3 | Lukas Fröhlich – Valentin Siebert | Available |
| $^{51}$V / $^{43}$Ca$_{shell}$ | Cage | 3 | Lukas Fröhlich – Valentin Siebert | Available |
| $^{55}$Mn / $^{43}$Ca$_{shell}$ | Cage | 3 | Lukas Fröhlich – Valentin Siebert | Available |
| $^{57}$Fe / $^{43}$Ca$_{shell}$ | Cage | 3 | Lukas Fröhlich – Valentin Siebert | Available |
| $^{88}$Sr / $^{43}$Ca$_{shell}$ | Cage | 3 | Lukas Fröhlich – Valentin Siebert | Available |
| $^{95}$Mo / $^{43}$Ca$_{shell}$ | Cage | 3 | Lukas Fröhlich – Valentin Siebert | Available |
| $^{97}$Mo / $^{43}$Ca$_{shell}$ | Cage | 3 | Lukas Fröhlich – Valentin Siebert | Available |
| $^{135}$Ba / $^{43}$Ca$_{shell}$ | Cage | 3 | Lukas Fröhlich – Valentin Siebert | Available |





| $^{137}$Ba / $^{43}$Ca$_{shell}$ | Cage | 3 | Lukas Fröhlich – Valentin Siebert | Available |
| $^{208}$Pb / $^{43}$Ca$_{shell}$ | Cage | 3 | Lukas Fröhlich – Valentin Siebert | Available |
| $^{238}$U / $^{43}$Ca$_{shell}$ | Cage | 3 | Lukas Fröhlich – Valentin Siebert | Available |