# Peer review of "HIPPO environmental monitoring: Impact of phytoplankton dynamics on water column chemistry and the sclerochronology of the king scallop (*Pecten maximus*) as a biogenic archive for past primary production reconstructions"

_Earth System Science Data, 2023_

## Referee Comment (RC2)

[referee-annotated manuscript omitted]

---

## Author Response (AR1)

Dear editor, dear reviewers,

To begin with, I want to express my gratitude for your feedbacks on our article entitled "HIPPO environmental monitoring: Impact of phytoplankton dynamics on water column chemistry and the sclerochronology of the king scallop (Pecten maximus) as a biogenic archive for past primary production reconstructions.". We have carefully considered all comments made by the reviewers and have made the required revisions to our article. In order to provide more detailed responses to the remarks, I will address them on behalf of all the authors, addressing Niels de Winter's comments first, followed by Andrew Johnson's.
* * *
**Reply to Niels de Winter**

First of all, I would like to thank you for your enthusiasm for this article and the broader concept of environmental monitoring.

In your comments, you state that it would be interesting to maintain this type of monitoring over several years to obtain the most representative results possible. However, such a survey necessitates substantial financial and human resources, as well as extensive time commitment. Preparations for this type of monitoring, including assembling various instruments, and subsequent data analysis took several months before, during and after the survey. Moreover, more than 20 people were involved in this monitoring, requiring intricate organization and coordination, which can be challenging to sustain over multiple years, especially considering each person's simultaneous involvement in their own projects. Nevertheless, it is worth mentioning that our study site, Lanvéoc, has been and is still subject to chemical and phytoplankton monitoring by the IUEM observatory since 2019, with bi-monthly expeditions. Regrettably, the observatory's measurements encompass a narrower range of parameters (excluding trace elements in seawater and shells for example), limiting the extent of analysis. However, we agreed with your perspective that it is unfortunate that our monitoring efforts were limited to a single year. Unfortunately, external constraints hindered us from conducting the study over several years.

Regarding your request to present element-to-calcium ratios in the shells compared to concentrations in seawater at the end of the publication, we believe this comparison is not relevant in the context of this article. Furthermore, the units of these two variables are inherently incomparable (one being an element-to-calcium ratio and the other representing chemical concentrations). Consequently, we have made the decision not to include these results in order to avoid making figure 10 and the accompanying text excessively long. We encourage readers and interested scientists to explore the data as they deem appropriate and manipulate it according to their own needs.

Finally, we have taken note of the few corrections you suggested for our manuscript and incorporated them accordingly.
* * *
**Reply to Andrew Johnson**

Just like Mr. De Winter, I would like to express my gratitude for your valuable feedback on our article and all the suggested modification you made on the text. We have incorporated all the corrections you requested in the PDF document you provided. We are pleased to hear that you were able to access the data on the SEANOE platform. Regarding the issue with the time format, I do not know whether it was an actual error on my part or simply a formatting problem related to the computer's language settings. Moreover, as requested, we have paid particular attention to the 'References' section, and have made the necessary corrections.

I will now address your main comment regarding the respect of the 'Completeness' criterion. We assure you that no data has been concealed or withheld for the purpose of writing other publications in the future. All the data generated during the 2021 environmental monitoring has been made available on the SEANOE platform. However, it was impossible to present results for every single parameter in this article. Therefore, we have chosen to display only the results that we deemed most relevant to our study objectives, providing readers with an understanding of the type of results achievable with this dataset. However, we have rephrased the sentence you highlighted in the initial version to avoid any further misinterpretation.
* * *
I hope that these comments answer the questions raised by Niels de Winter and Andrew Johnson and I remain at your disposal for further information. We are confident that this important work significantly improved our manuscript and we would like to thank you as well as the reviewers for your involvement in the enhancement of our paper. The revised manuscript is ready to be submitted.

Yours sincerely,

Valentin Siebert, on behalf of all the authors